# Unidirectional chiral scattering from single enantiomeric plasmonic nanoparticles

Yuanyang Xie [1,2] ✉, Alexey V. Krasavin[1,2], Diane J. Roth [1] & Anatoly V. Zayats [1] ✉

Controlling scattering and routing of chiral light at the nanoscale is important for optical information processing and imaging, quantum technologies as well as optical manipulation. Here, we introduce a concept of rotating chiral dipoles in order to achieve unidirectional chiral scattering. Implementing this concept by engineering multipole excitations in helicoidal plasmonic nanoparticles, we experimentally demonstrate enantio-sensitive and highly-directional forward scattering of circularly polarised light. The intensity of this highly-directional scattering is defined by the mutual relation between the handedness of the incident light and the chirality of the structure. The concept of rotating chiral dipoles offers numerous opportunities for engineering scattering from chiral nanostructures and optical nano-antennas paving the way for innovative designs and applications of chiral light-matter interactions.

Chirality describes a geometrical property of an object which cannot be superimposed with its mirror image using any combination of translation and rotation operations. Such geometrical characteristics, translated into enantio-sensitive optical and chemical properties, are of great importance in the physics of light-matter interactions, chiral chemistry and catalysis, as well as various areas of bioscience[1–3]. The most well-studied representations of material chirality in optics are circular birefringence and circular dichroism, resulting from different real and imaginary parts of the refractive index of the material, respectively, for circularly polarised light of opposite handedness[4,5].

Phenomenologically, on a microscopic level, chiral optical properties are related to the excitation of collinear electric and magnetic dipoles, while the phase between them determines the handedness of the emitted or scattered light. Nanotechnology has made a game-changing impact on the field of optical chirality providing tools to create artificial nanostructured materials with engineered or enhanced chiral properties[6–9]. In particular, the development of nanostructures with a designed enhanced chiral response leads to the enantio-, spatially- and spectrally-selective control of optical, chemical or biological processes at the nanoscale[7,10–14]. This offers a rigorous means for manipulation of chiral light, which can be used not only in modern optoelectronic components, for data routing, processing and storage[15–17], but also presents a key technology for telecom devices based on quantum communication and encryption[18,19]. No less important applications of the enhanced chiral response at the nanoscale are sensing and active control of enantio-selective molecular processes, which plays a fundamental role in biology and medicine[20–23].

The control of directionality and polarisation state of light wavefronts is one of the central questions in conventional and quantum photonic applications[24,25]. Particularly, it is a key component in the design of chiral light-matter interactions. While its importance is quite obvious for routing the chiral light[15,26–29] and for sensing in order to increase the collection efficiency, it has been also used for engineering photonic spin-orbit interactions[30], directing chiral Raman signals[31] and single particle detection and sizing[32]. The interplay between chirality and directionality of light emission also lays a basis for a new field of chiral quantum optics[33,34]. In this respect, various types of chiral directionally-selective nanostructures have been proposed[15,16,30,32,35]. The experimental work on directionality is supported by the theoretical understanding trying to identify the sets of elemental modes required for the directional light scattering or emission. Rotating (circularly polarised) electric dipoles as well as Huygens and Janus dipolar sources[36–38] have been developed for engineering counter-intuitive unidirectional coupling to plasmonic and dielectric

[1]Department of Physics and London Centre for Nanotechnology, King's College London, London WS2R 2LS, UK. [2]These authors contributed equally: Yuanyang Xie, Alexey V. Krasavin. ✉e-mail: yuanyang.xie@kcl.ac.uk; a.zayats@kcl.ac.uk

waveguides[39] and far-field directional scattering[40], harnessing the interplay between orthogonal electric/electric, magnetic/magnetic and electric/magnetic dipoles through near-field interference. Chiral dipoles can be constructed with an out-of-phase combination of collinear electric and magnetic dipoles with the properly balanced amplitudes[36]. While such dipolar sources emit pure circularly polarised light, offering enantio-sensitive coupling to molecules or nanostructures, they have a trivial dipolar-like emission pattern. The respective fields of electric and magnetic components of such dipolar sources are orthogonal to each other and do not interfere, making it impossible to exploit the near-field interference to achieve directional properties.

In this work, we develop a concept of rotating chiral dipoles to achieve highly-directional and enantio-sensitive scattering into purely circularly polarised light states, with the directionality controlled by the spin of the dipoles. We experimentally demonstrate this effect by engineering the phase-resolved excitation of electric and magnetic dipoles in chiral plasmonic nanohelicoids using single-particle Fourier microscopy. The observed enantiomer-sensitive behaviour, which depends on the relation between the chirality of the nanoparticle and the handedness of the incident light, is important in designing and controlling polarisation-sensitive nanophotonics, enantiomeric optical forces, detection and sorting of chiral objects, as well as chiral photocatalysis.

## Results

### Concept of a rotating chiral dipole

Complementing electric and magnetic dipoles, chiral $\sigma^{\pm}$-dipoles emitting light with pure circular polarisation can be introduced, where " $+$ " and " $-$ " correspond to right circular polarised (RCP) and left circularly polarised (LCP) emission, respectively[36]. They can be represented within a dipolar framework as a sum of co-directed electric $|\mathbf{p}\rangle$ and a magnetic $|\mathbf{m}\rangle$ dipoles with a certain ratio of the magnitudes $p = m/c$ and a $\mp \pi/2$ phase shift (Fig. 1a, b)[36], for $\sigma^{\pm}$-dipoles, respectively. Right- $|\sigma_x^+\rangle$ and left- $|\sigma_x^-\rangle$ handed chiral dipoles directed along the $x$-axis are given by $|\sigma_x^{\pm}\rangle = |\mathbf{m}_x\rangle \pm i |\mathbf{p}_x\rangle = (1,0,0)|\sigma^{\pm}\rangle = (1,0,0)|\mathbf{m}\rangle \pm i(1,0,0)|\mathbf{p}\rangle$. Chiral dipoles radiate purely circularly polarised light in the direction normal to the dipole orientation with a typical dipolar radiation pattern (Fig. 1a, b). In contrast, a rotating electric dipole radiates circularly polarised light of opposite handedness in the opposite directions (Supplementary Fig. 1a).

In a direct analogy to rotating electric dipoles, two perpendicular chiral dipoles with a $\pm \pi/2$ phase shift generate a rotating chiral dipole (see Supplementary Fig. 6 for details). Right-handed ( $+$ ) chiral dipole rotating clockwise ( $+$ ) or counterclockwise ( $-$ ) in the $xy$-plane and hence directed along the $z$-axis is given by $|\sigma_z^{+\pm}\rangle = (1, \pm i, 0)|\sigma^+\rangle$, where the axis of rotation and the rotation direction are marked by the subscript and the second sign in the superscript, respectively, and $\exp(-i\omega t)$ time dependence is assumed. The emission from the rotating chiral dipoles retains its pure handedness, but it is unidirectional with the direction depending on the rotation (spin vector) direction (Fig. 1c–f). If the handedness of a chiral dipole is changed to the opposite while keeping its spin the same, the radiation direction is changed to the opposite. In all cases, the emission is predominantly perpendicular to the rotation plane. Mnemonically, the handedness of the chiral dipole, its rotation direction and the emission direction obey the hand rule: with the left or right hand chosen to match the handedness of the chiral dipole and the fingers bent to indicate the direction of the rotation, the thumb will show the direction of the radiation. Mathematically, this can be expressed as $\mathbf{l} = \pm \mathbf{s}$, where $\mathbf{l}$ marks the direction of radiation, $\mathbf{s}$ is the spin vector of the rotating chiral dipole, related to the rotation direction by the right-hand rule, and the choice of $+$ or $-$ is defined by the handedness of the dipole. Consequently, for right-handed rotating chiral dipoles the emission is along their spin, and for left-handed counterparts opposite. In the case of chiral dipoles

with "elliptical" rotation (the combination of elliptically polarised magnetic and electric dipoles), the ratio of the forward and backward scattered intensities can be controlled by the phase difference between the $x$- and $y$- chiral dipole components, with equal intensities achieved in the case of a linear chiral dipole when the components are in phase (Fig. 1g).

The introduced concept of rotating chiral dipoles can be linked to other common electromagnetic dipolar sources (Supplementary Fig. 6). Generally, a pair of electric and magnetic dipoles or a pair of chiral dipoles of different handedness forms a complete basis for representing any dipolar state along a given direction: $|\mathbf{d}\rangle = \mathbf{p}|\mathbf{p}\rangle + \mathbf{m}|\mathbf{m}\rangle = \sigma^+|\sigma^+\rangle + \sigma^-|\sigma^-\rangle$, where $\mathbf{p} = (p_x, p_y, p_z)$ and similarly for other amplitude vectors involved. A basis of four chiral dipoles or four rotating chiral dipoles is needed to produce an arbitrary in-plane dipolar state, while six states from either of these two bases form a complete set for the description of any dipolar state in three dimensions (see Supplementary Note 1 for details). For example, the rotating electric (magnetic) dipoles can be represented as two rotating out-of-phase (in-phase) chiral dipoles with the same spin and opposite handedness, while a Huygens dipole, unidirectionally emitting a linearly polarised light (Supplementary Fig. 1b), can be viewed as two rotating chiral dipoles with opposite spins and handedness.

### Rotating chiral dipoles in plasmonic nanohelicoids

Among various realisations of chiral nanoparticles[8,9,11,12,14,41], plasmonic nanohelicoids (Fig. 2a) provide strong resonant chiroptical effects[6,7,10,13,42,43]. They can be either right-handed (D-nanohelicoids) or left-handed (L-nanohelicoids)[42,43], which is defined by the handedness of glutathione used for their fabrication (see Methods for the fabrication details). Broad spectral tuneability of their resonant response is favourable for designing electric and magnetic dipolar resonances with the required amplitudes and relative phase, which makes them particularly suitable for the realisation of chiral dipoles.

At the wavelength of the magnetic dipolar resonance, the optical properties of L-nanohelicoids of a 180 nm size in a $SiO_2$ matrix considered here are governed by the electric and magnetic dipolar responses (see Supplementary Note 2 for the details of the multipole decomposition), showing only a minor contribution from an electric quadrupole and no contribution from its magnetic counterpart (Supplementary Fig. 7b)). The contribution from the magnetic dipole is strongest if the handedness of the excitation light and the nanoparticle is the same, and suppressed in the opposite case. Particularly, under illumination with a $x$-polarised plane wave propagating in the $z$-direction, the dominant contributions come from an electric dipole with a magnitude $|p_x|$ parallel to the direction of the incident electric field and a magnetic dipole with a magnitude $|m_y|/c$ perpendicular to it (Fig. 2b). Near the magnetic resonance at a wavelength of around 620 nm, their magnitudes are close to each other. Simultaneously, the phase between them is $\pi$, i.e., they are out of phase (Fig. 2c). These conditions correspond to the excitation of a Huygens dipole at this wavelength. There are no electric or magnetic dipole components excited along the propagation direction of the incident wave ($z$-direction). Importantly, there exist also a magnetic dipole with a magnitude $|m_x|$ parallel to the dominant $|p_x|$ electric counterpart and an electric dipole with a magnitude $|p_y|$ parallel to the dominant $|m_y|$ magnetic counterpart. Furthermore, the phase shift between $p_x$ and $m_x$ as well as between $p_y$ and $m_y$ at the resonant wavelength of 620 nm is $-\pi/2$ (see dashed blue and red curves in Fig. 2c, respectively), which signifies the formation of two $|\sigma^-\rangle$ chiral dipoles along the $x$- and $y$- directions and their dominance over their $|\sigma^+\rangle$ counterparts. Such predominance is the origin of the overall chiroptical response related to the geometrical chirality of the nanohelicoid. This can be seen directly by decomposition of the excited dipolar state in the chiral dipole basis $\sigma^{\pm} = (\mathbf{m}/c \mp i\mathbf{p})/2$ (see Supplementary Note 3 for details) as a difference in the excitation of right- and left-handed chiral dipoles. Comparing $|\sigma_x^-|$ with $|\sigma_x^+|$, and

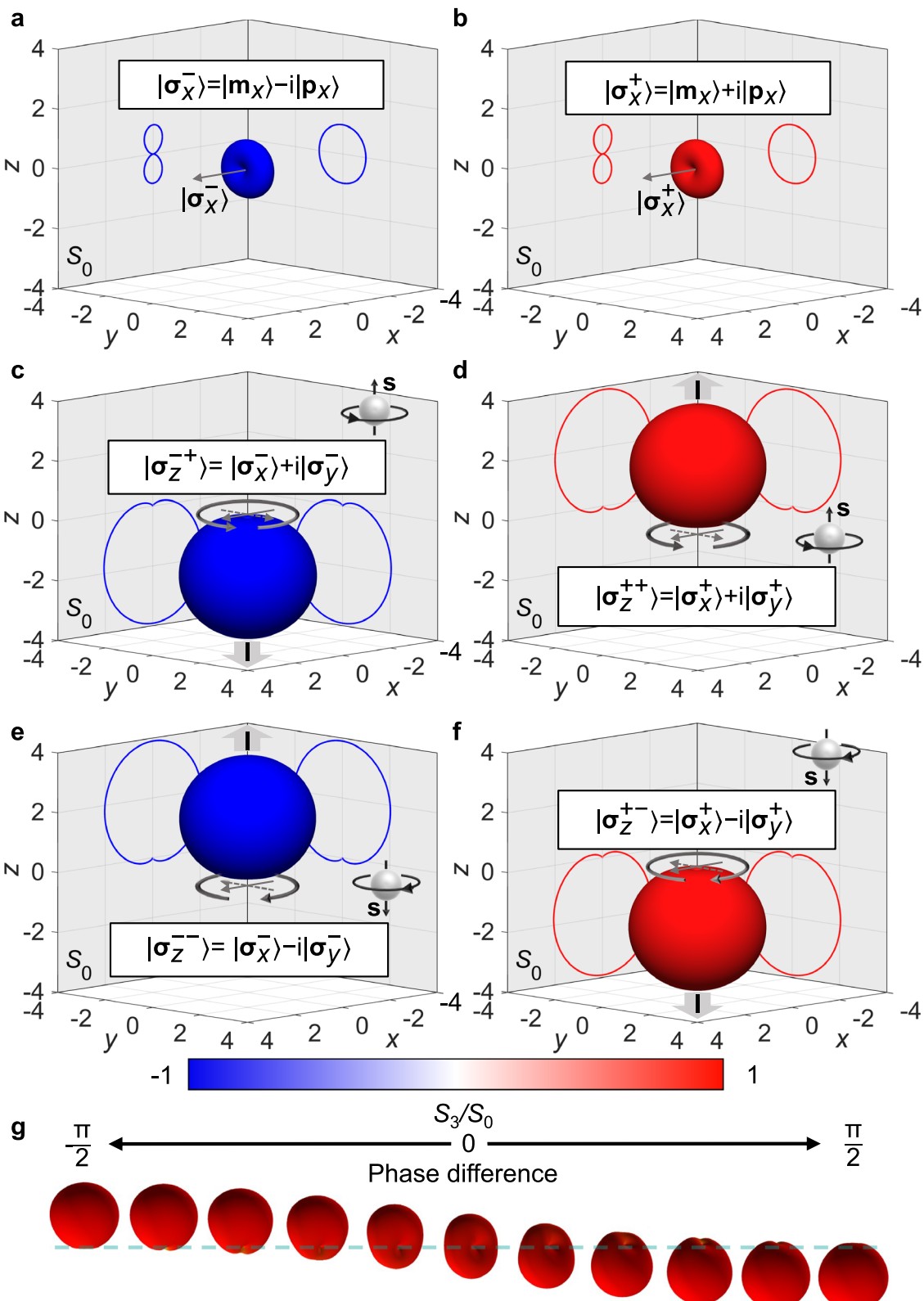

**Fig. 1 | Radiation patterns of chiral dipoles.** Far-field radiation intensity and polarisation diagrams of (**a**) left-handed $|\sigma_x^-\rangle$ and (**b**) right-handed $|\sigma_x^+\rangle$ chiral dipoles, emitting pure RCP and LCP light, respectively. The arrows show the direction of the chiral dipole. **c–f** Far-field radiation intensity diagrams of the rotating $|\sigma_z^{\pm\pm}\rangle$ chiral dipoles with indicated handednesses and rotation directions. The colour map shows $S_3/S_0$ Stokes parameter which represents the handedness of the emitted light (values 1 and −1 correspond to RCP and LCP, respectively, defined from the point of view of the source). The arrows show the rotation direction of the chiral dipole. **g** Far-field radiation intensity and polarisation diagrams of right-handed "elliptically" polarised rotating chiral dipoles with varied phase difference between equal $x$- and $y$- components. The phase step is $\pi/10$.

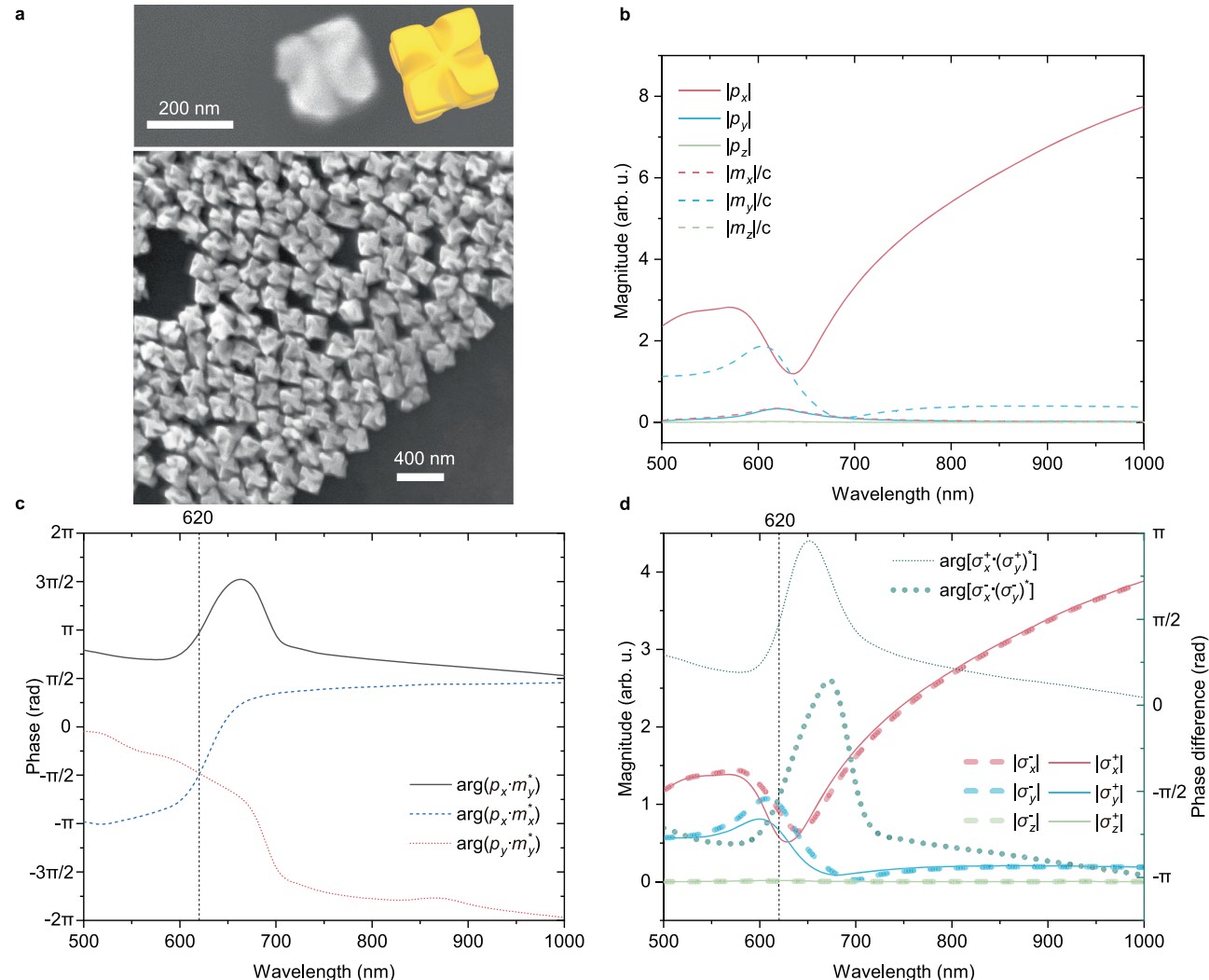

**Fig. 2 | Optical properties of plasmonic L-nanohelicoids. a** Scanning electron microscopy images (top view) of an individual gold L-nanohelicoid (schematics is shown in yellow) and a L-nanohelicoid assembly on a substrate showing the uniformity of the fabricated nanostructures; the average size of the nanohelicoids is around 180 nm. **b**–**d** Numerically calculated (**b**) magnitudes and (**c**) relative phases of the electric and magnetic dipoles excited in a 180 nm gold L-nanohelicoid in a SiO₂ matrix with a x-polarised plane wave propagating in the −z direction, and (**d**) corresponding magnitudes and phases of the excited chiral dipoles. Source data are provided as a Source Data file.

---

$|\sigma_y^-|$ with $|\sigma_y^+|$ presented in Fig. 2d one can see that

$$|\sigma_x^-| \approx |\sigma_y^-| > |\sigma_x^+| \approx |\sigma_y^+|, \qquad (1)$$

which marks the dominant excitation of $|\sigma^-\rangle$ and the appearance of chiroptical effects. Furthermore, in the rotating chiral dipole basis, the relative phases between $|\sigma^\pm\rangle$ components along the x- and y- axes $\arg\left[\sigma_x^+ \cdot (\sigma_y^+)^*\right] = \pi/2$ and $\arg\left[\sigma_x^- \cdot (\sigma_y^-)^*\right] = -\pi/2$ signify the excitation of both $|\sigma_z^{+-}\rangle$ and $|\sigma_z^{-+}\rangle$ rotating chiral dipoles under linearly polarised excitation, but with different magnitudes defined by the chirality of the object.

Under circularly polarised illumination, a second set of electric and magnetic dipoles (orthogonal counterparts of that presented in Fig. 2b) will be excited with a phase delay determined by the handedness of the incident light. Generally, this will result in a sum of elliptically polarised electric and magnetic dipoles with certain elliptical trajectories and phase delays. However, at a resonant wavelength of 620 nm light, $|\sigma_z^{-+}\rangle$ is excited in the case of LCP illumination and $|\sigma_z^{+-}\rangle$ in the case of RCP, because of balanced excitation of electric and magnetic dipoles in nanohelicoids (Fig. 2b): for LCP (RCP) illumination

$|\boldsymbol{\sigma}_x^-\rangle$ and $|\boldsymbol{\sigma}_y^-\rangle$ ($|\boldsymbol{\sigma}_x^+\rangle$ and $|\boldsymbol{\sigma}_y^+\rangle$) are predominantly excited, importantly with the same magnitudes and a $-\pi/2$ ($\pi/2$) phase difference, resulting in the rotation direction coinciding with the handedness of the incident field (Fig. 3b, c). Thus, the direction of the spin of the excited rotating chiral dipoles is determined by the handedness and the propagation direction of the illuminating light. This can be further proven analytically (see Supplementary Note 4 for details). At the same time, the magnitudes of the excited $|\sigma^-\rangle$ and $|\sigma^+\rangle$ chiral dipoles (as can be seen from Eq. (1) for x-polarised excitation), and therefore the magnitudes of the resulting rotating chiral dipoles are unequal for the LCP and RCP, which signifies the interaction of the chirality of the illuminating light with the chirality of the nanohelicoid, i.e., the enantiosensitivity and chiral nature of the scattering process.

Nanosphere and nanocube particles of the same size show predominantly electric dipolar and electric quadrupolar optical responses, with only a small contribution of magnetic dipoles (cf. Supplementary Figs. 2, 3 and 7), which hinders the excitation of pure rotating chiral dipole states. A certain directionality of the scattering in the case of the nanocubes is the result of favourable interference of electric dipolar and electric quadrupolar scattering[38]. However, as

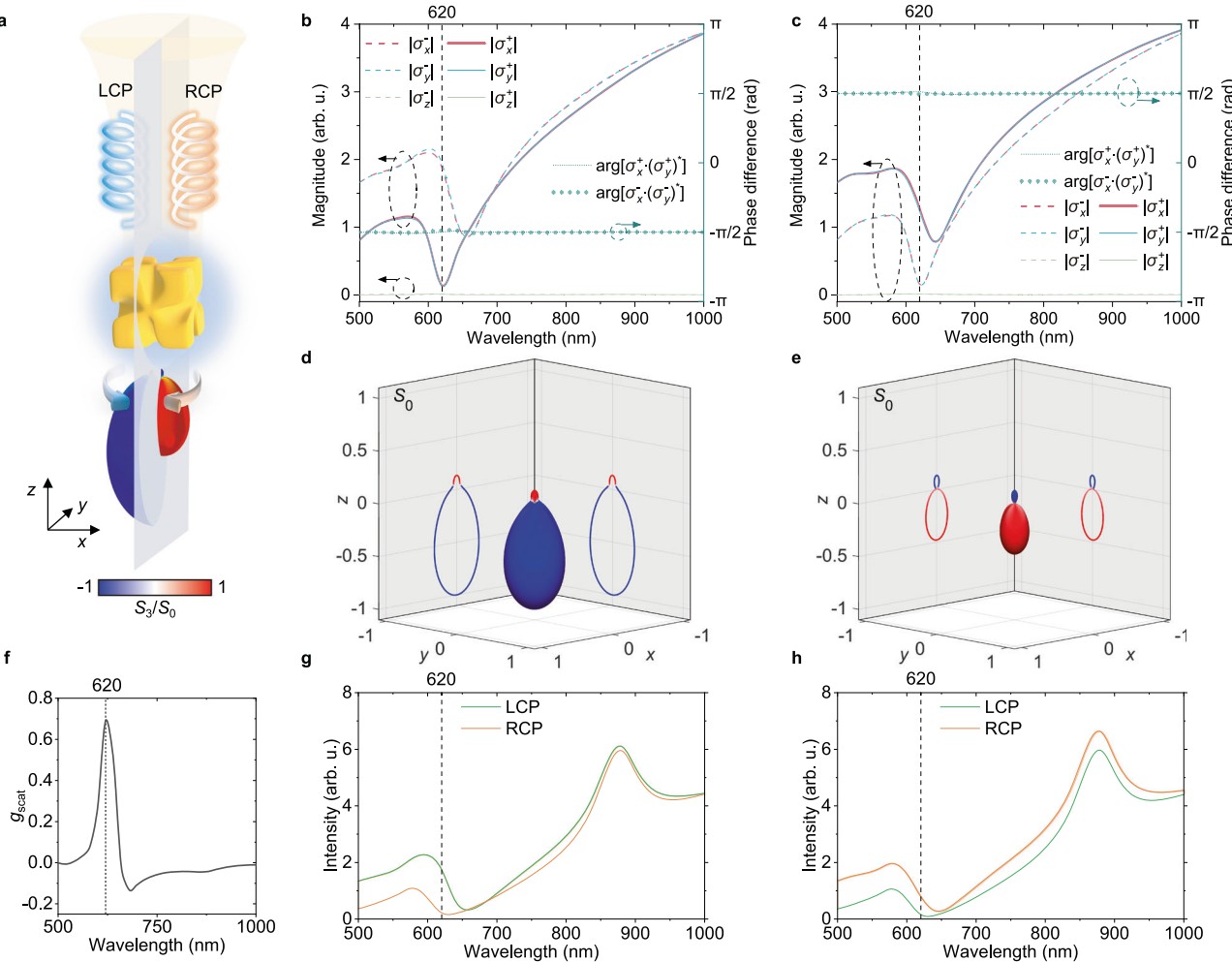

**Fig. 3 | Enantio-sensitive excitation of rotating chiral dipoles by circularly polarised light. a** The schematics of the excitation of rotating chiral dipoles in a L-nanohelicoid by circularly polarised light with the corresponding simulated far-field scattering intensity and polarisation diagrams. **b**, **c** Numerically calculated magnitudes and relative phases of the chiral dipoles excited in a 180 nm gold L-nanohelicoid in SiO₂ with (**b**) a LCP and (**c**) a RCP plane wave propagating in the $-z$ direction. **d**, **e** Far-field scattering intensity and polarisation diagrams calculated at the wavelength of 620 nm in cases (**b**) and (**c**), respectively. The colour indicates the polarisation state of the scattered light calculated from Stokes parameters $S_3/S_0$ (1: RCP; −1: LCP). **f** Simulated spectral dependence of a nanohelicoid scattering $g$-factor. **g**, **h** Integrated intensities for RCP and LCP polarisations of the scattered light in cases (**b**) and (**c**), respectively. Source data are provided as a Source Data file.

these nanoparticles are achiral, they interact with light of different circular polarisations in the same fashion.

From the radiation patterns of the ideal rotating chiral dipoles (Fig. 1c–f), one can see that the ratio between the magnitudes of the excited rotating chiral dipoles defines the balance between the forward scattering with the same polarisation as the incident one and backward scattering with the opposite handedness. As this ratio is wavelength-dependent (Fig. 3b, c), gradual spectral evolution of the scattering pattern is observed. Near the magnetic resonance at a wavelength of 620 nm (Fig. 3b, c), one dominant rotating chiral dipole is excited, while the amplitude of a (secondary) rotating chiral dipole with the opposite handedness is minimised. The 180-nm nanohelicoid size chosen in the study offers the highest contrast between the opposite handednesses (Supplementary Fig. 8). The resulting emission directionality is similar to the ideal case shown in Fig. 1c, f, revealing vastly predominant forward scattering with the same handedness as the illuminating light (Fig. 3d, e). At the same time, since a nanohelicoid is a chiral object, it interacts differently with circularly polarised light of different handedness. Particularly, for a given chirality of the nanohelicoid, RCP and LCP light is forward-scattered with unequal efficiency. When the direction of the rotation of the electric field of the

incident wave coincides with the direction of the nanohelicoid twist (spin of the excited rotating chiral dipole and chirality of the nano-helicoid are aligned), forward scattering is stronger. Both effects, the dominant scattering of the light of the same handedness as the incident light and the role of nanohelicoid chirality (cf. LCP in Fig. 3g and RCP in Fig. 3h) can be observed in the total emitted intensities (Fig. 3g, h). It is also interesting to note that the chiroptical response expressed as a conventional scattering $g$-factor

$$g_{\text{scat}} = 2\,\frac{C_{\text{scat}}^{\text{LCP}} - C_{\text{scat}}^{\text{RCP}}}{C_{\text{scat}}^{\text{LCP}} + C_{\text{scat}}^{\text{RCP}}}, \tag{2}$$

where $C_{\text{scat}}^{\text{LCP}}$ and $C_{\text{scat}}^{\text{RCP}}$ are the scattering cross-sections for LCP and RCP light (Supplementary Fig. 9), respectively, reaches its maximum at the 620 nm wavelength at which pure chiral dipoles are excited (Fig. 3f). The same can be observed for an extinction $g$-factor (Supplementary Fig. 10). This gives a confirmation that the efficiency of the excitation of pure rotating chiral dipolar states is influenced by the presence of structural chirality. The scattered fields can be also affected by the excitation of higher-order multipoles, e.g., the peak just above 860 nm corresponds to the electric quadrupolar resonance

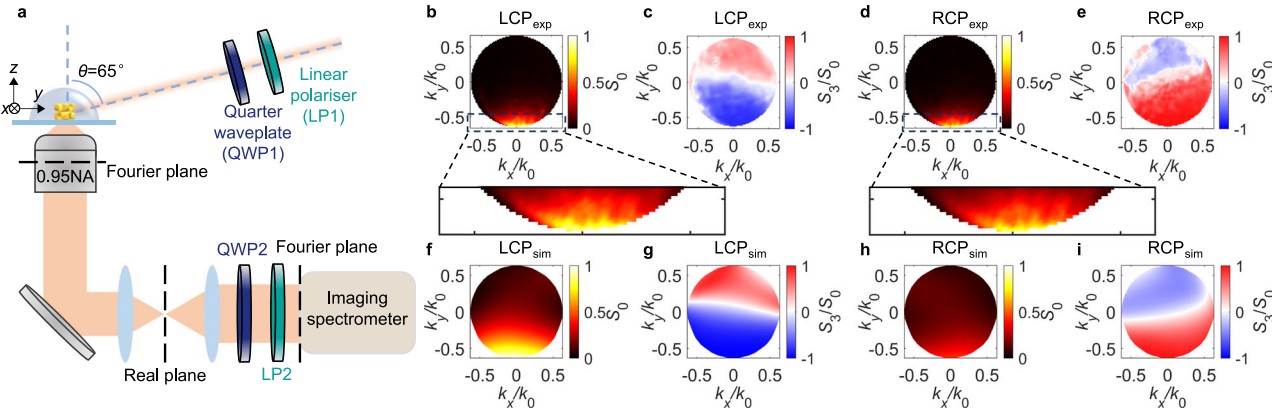

**Fig. 4 | Experimental observation of chiral scattering from single L-nanohelicoid. a** Schematic of the experimental setup. **b–e** Experimental and **f–i** numerical Fourier-space maps of normalised $S_0$ (**b, f, h**) showing the intensity of the scattered light and $S_3/S_0$ (**c, g, e, i**) showing its polarisation for (**b, c, f, g**) LCP and (**d, h, e, i**) RCP illumination. In both experiments and numerical simulations the scattering intensity in any measured direction was much larger than the signal noise (numerical noise in the case of simulations) to give reliable valies for $S_3/S_0$. In the simulations, the nanohelocoid was rotated by 45° to better represent the experiment.

(Supplementary Fig. 11). Making an impact on the polarisation state of the scattered light, the latter can smear-out the contribution from the pure chiral dipole, resulting in the decrease of the $g$-factors. Note that comparing Supplementary Fig. 7a and c, one can see that the contributions of the excited magnetic dipole and excited electric dipole are the same when the handedness of the illuminating light matches the handedness of the nanohelicoid (at the same time, for the opposite handedness, the contribution of a magnetic dipole is reduced). Consequently, a well-defined rotating chiral dipole is excited, not only providing a larger scattering intensity, but also a better directionality (cf. Fig. 3d and e) and a better LCP/RCP contrast (cf. Fig. 3f and g). As can be expected, nanohelicoids with the opposite handedness (D-nanohelicoids) behave in a reverse manner, as can be seen from their scattering diagrams presented in Supplementary Fig. 12. It should be noted that the chirality of the nanoparticle response is determined by the presence and strength of the aligned and spectrally overlapped electric and magnetic dipoles (i.e., the strength of the resulting chiral dipole). As for the chirality of light, in the case of the excitation of pure chiral dipoles shown in Fig. 1, it does not depend on whether the dipole is a conventional one or rotating, it is either pure RCP or pure LCP, but the directionality is different in these two cases. Overall, rotating chiral dipoles possess all the properties of a distinct modal set for dipolar excitations: 1) pure chiral dipolar states can be excited, e.g., in nanohelicoids as shown above, or indeed in other plasmonic nanoparticles satisfying the resonant conditions, 2) the set is complete, i.e., any dipolar plasmonic excitation can be decomposed in these modes.

**Experimental demonstration**

For experimental demonstration, gold nanohelicoids with a size of 180 nm were prepared by an amino-guided growth method[6] (see Methods). They were deposited on a silica coverslip by spin coating, during which the nanohelicoids were well separated to take measurements on individual particles (Supplementary Fig. 13). The nanohelicoids were further covered with a $SiO_2$ layer to provide a uniform surrounding environment. Angle-resolved scattering from a single nanohelicoid for both circular polarisations were measured using a Fourier plane imaging technique with 610 nm illumination (Fig. 4a) (see Methods for the details of the experimental set-up). The illumination wavelength was chosen to match the maximum of the experimentally measured circular dichroism of the nanohelicoids in $SiO_2$ (Supplementary Fig. 14). The maps of $S_0$ and $S_3$ plotted for both LCP and RCP polarisations of

the incident beam and representing the intensity and polarisation of the scattered light, respectively, show predominant forward scattering with a high-quality polarisation state with the same handedness as that of the incident light (Fig. 4b–e), confirming the numerical predictions and the analysis based on the excitation of rotating chiral dipoles (Fig. 4f–i). Importantly, compared to the scattering of RCP incident light, the LCP illumination results in stronger forward scattering, signifying the chiral nature of the scattering. The relative difference of the scattered intensities of LCP and RCP light in the zoomed-in area $2(I_{LCP} - I_{RCP})/(I_{LCP} + I_{RCP})$ corresponds to $\approx 15\%$, which is lower than the theoretical prediction of $\approx 50\%$, due to the deviations of the helicoid shapes from the ideal modelled one, resulting in the breaking of the balance between matched amplitudes of the excited electric and magnetic dipoles at the resonant wavelength.

In the single-particle scattering measurements, the orientation of the nanohelicoid with regard to the laser beam is generally unknown. To take this into account, numerical simulations for characteristic spatial orientations of the nanohelicoid with respect to the direction of the illumination light were performed (Supplementary Fig. 4). It was found that although there are some variations in the far-field scattering patterns due to a non-ideal balance of the magnitudes and phases of the excited electric and magnetic dipoles, the handedness of the forward scattered light and the ratio between scattered intensities of LCP and RCP light remain largely the same, which explains the robustness of the experimental results. The variation of the permittivity of the surroundings and/or the presence of a substrate with dissimilar optical properties affects the excitation of the plasmonic resonances and hence the purity of the excited rotating chiral dipole state. However, as the numerical modelling shows, the key phenomenon of the directional enantio-sensitive scattering can still be observed (Supplementary Fig. 15).

## Discussion

We have introduced a concept of rotating chiral dipoles for achieving directional enantio-sensitive scattering from chiral nanoparticles. The scattering direction, handedness of scattered light and chiral dipole rotation are linked through the hand rule, the choice of the left or right hand is determined by the chirality of the dipole. Such dipoles can be excited in nanoparticles by circularly polarised light with both handedness of the chiral dipole and its rotation spin determined by the spin of the illuminating light. Furthermore, the introduced rotating chiral dipoles produce a

complete set into which any dipolar excitation can be decomposed. The observations of the circularly polarised light scattering from a single plasmonic chiral nanoparticle reveal the excitation of pure rotating chiral dipole states, leading to vastly predominant forward scattering with the same handedness as the illuminating light. The experimentally observed scattering is enantio-sensitive, confirming the theoretical predictions and demonstrating the chiral character of the optical interaction. Particularly, the scattering is enhanced if the chirality of the illuminating light and the nanoparticle are the same. Due to the amplitude and phase differences of the electric and magnetic dipoles of the nanohelicoid compared to those required for an ideal rotating chiral dipole, the scattering of light with the handedness opposite to that of the nanohelicoid is suppressed, but present in the same directional manner. The obtained results can be useful for enantio-sensitive optomechanical manipulation[44–46] (see Supplementary Note 7 for discussions of enantio-sensitive optical forces), directionality-assisted increase of the efficiency of enentio-selective sensing[7] and chiral photoluminescence[47,48], as well as quantum sensing[49].

## Methods

### Materials
Hexadecyltrimethylammonium bromide (CTAB, ≥98%), hexadecyl-trimethylammonium chloride (CTAC, ≥98%), L-glutathione (≥98%), Au(III) chloridetrihydrate (HAuCl$_4$ · 3H$_2$O, 99.9%), sodium borohydride (NaBH$_4$, 99.99%), potassium iodide (KI, ≥99.5%) and L-ascorbic acid (AA, ≥99.0%) were purchased from Sigma-Aldrich. The water with a resistivity of 18.2 MΩ cm$^{-1}$ (DI water) was used throughout the procedures.

### Fabrication of plasmonic nanohelicoids
The L-helicoid nanoparticles were grown from octahedral gold seeds using a method described in ref. 6. The helicoid growth solution included 29.04 mL DI water, 6.4 mL 0.1 M CTAB and 640 μL 10 mM HAuCl$_4$. Then, a 3.6 mL 0.1 M ascorbic acid solution was injected into the growth solution as a reducing agent. Then, the chiral growth was triggered by adding 16 μL of 5 μM L-glutathione to a 320 μL solution of the octahedral seeds which was prepared by a two-step growth method[50]. After 2 h in a water bath at 30 °C, the particles were washed and collected by centrifuging (340×$g$, 5 min) 3 times and kept in a 1 mM CTAB solution for further use.

The helicoid nanoparticles embedded in SiO$_2$ were prepared in two steps. First, helicoid nanoparticles were dispersed on a plasma-cleaned glass by spin coating. Then, a SiO$_2$ layer was deposited on the top of helicoid nanoparticles by a sol-gel spin coating method[51]. Briefly, a SiO$_2$ sol-gel solution was prepared mixing 223 μL of TEOS, 992 mg of isopropanol, 278 μL of DI water and 12.46 μL of a 37% HCl solution. The mixture was kept stirring for 80 min in a water bath at 70 °C before the spin coating.

### Finite element method numerical simulations
Scattering of light from a single helicoidal plasmonic nanoparticle was simulated using the finite element method (COMSOL Multiphysics software) in a scattered field formulation. The nanohelicoid was positioned with its major axes aligned with the simulation reference frame and the wave vector of the illumination light wave was set in the $-z$ direction (see Fig. 3 for the co-ordinate axis orientation). The dielectric permittivity of gold, water and SiO$_2$ were taken from the tabulated experimental data[52–54]. The simulation domain was surrounded by a perfectly matching layer to assure the absence of back-reflection of the scattered waves.

To calculate Stokes parameters, far-field components $E_\theta$ and $E_\phi$ of the radiated or scattered electric field were analysed. Stokes parameters $S_0$ and $S_3$ describing the total intensity and polarisation state of the field, respectively, were obtained in a usual way as $S_0 = |E_\theta|^2 + |E_\phi|^2$ and $S_3 = -2\,\mathrm{Im}(E_\theta \cdot E_\phi^*)$.

### Fourier microscopy measurements
The sample was illuminated by a collimated white light from a supercontinuum laser (NKT Photonics SuperK-EVO-HP), filtered at the wavelength of interest, and circularly polarised using a broadband linear polariser and a quarter waveplate. The illumination was performed through a hemispherical lens to avoid the change on the incident light polarisation, while index-matching oil was used between the lens and the sample to obtain the index-matching illumination conditions. The angle of incidence of the beam on the nanohelicoid was 65°. Such an illumination angle allows the observation of both forward and backward scattering in the implemented Fourier setup. The scattered signal from the nanohelicoid was collected by a 100X microscope objective with NA = 0.95, allowing collection within a large solid angle. Even at the largest collection angle of ≈72°, the change of the forward-scattered circular polarisation due to the transmission difference for its $s$- and $p$-polarised components at the glass/air interface is less than 1.2%, ensuring negligible variation of the polarisation state of the transmitted light. The Fourier plane (back focal plane of the detection objective) was then imaged onto a CCD camera using a set of relay lenses. The polarisation of the scattered signal was analysed using a quarter waveplate and a linear polariser, allowing the reconstruction of the Stokes parameters.

## Data availability
The data that support the findings of this study are available from the corresponding authors upon request. Source data are provided with this paper.

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

## Acknowledgements

A.V.K and A.V.Z. acknowledge support from the ERC iCOMM project (789340) and the UK EPSRC project (EP/W017075/1). Y.X. acknowledges support from China Scholarship Council. The authors are grateful to Prof. Ki Tae Nam for providing a CAD geometry of the nanohelicoid (432 helicoid III) and to Dr. Vittorio Aita for help with the experimental setup.

## Author contributions

Y.X., A.V.K. and A.V.Z. developed the concept, Y.X. and A.V.K. performed the simulations, Y.X. fabricated nanoparticles, Y.X. and D.J.R. performed the measurements, all authors contributed to the writing of the manuscript.

## Competing interests

The authors declare no competing interests.
