## [Transparent Peer Review file · Nature Communications]

Unidirectional chiral scattering from single enantiomeric plasmonic nanoparticles

Corresponding Author: Mr Yuanyang Xie

Version 0:

Reviewer comments:

Reviewer #1

(Remarks to the Author)

In the manuscript by Xie et al, the authors first introduce a new basis of “rotating chiral dipoles” and then explore the interaction between circularly polarized light and chiral plasmonic nanohelicoids. By engineering the amplitude and phase relationships between electric and magnetic dipoles within these structures, they have achieved controlled, directional scattering of light, which is sensitive to the chirality of both the incident light and the nanoparticle. After simulating these processes, the authors used Fourier microscopy to experimentally observe the chiral scattering from the nanohelicoids, confirming their numerical results. The study appears to be complete, but several points remain unclear and confusing. Overall, I feel the novelty of this paper is more technical than what the readers of Nature Communications might expect, and thus would fit a more specialized journal with proper revision. I will elaborate on these points below:

1. In terms of the “rotating chiral dipoles” concept, there is a statement “The basis of the four rotating chiral dipoles forms a complete set describing any dipolar source arbitrarily oriented in space.” This is clearly inaccurate since SI showed the set can only represent p_x , p_y , m_x , m_y , not p_z and m_z . I believe the set can be completed by adding two more rotating chiral dipoles, but it will probably destroy the elegant symmetry presented in the current set.
2. Now if we ignore p_z and m_z , then there is no fundamental difference between the “rotating chiral dipole” set and the linear p and m set. This is not a problem since the latter is known to be complete, since the introduction of a new set would still be justified if the authors can offer a real physical object that is at least theoretically a pure base in the new set. Ideally, there should be some simple symmetry argument showing this object must be a pure rotating chiral dipole. Alternatively, there is some new property demonstrated by the pure rotating chiral dipole. The structure studied in this manuscript satisfied neither. As the manuscript introduced, directional circularly polarized scattering can be engineered using other frameworks too. It doesn't present any physical insight if one just defines an arbitrary orthogonal basis set, and for an arbitrary structure finds coincidentally a condition that makes the optical response similar to one of the bases.
3. Another vague statement is “In contrast, for chiral dipolar sources, in the radiation patterns of the involved collinear magnetic and electric dipoles, the respective electric fields are orthogonal to each other and do not interfere, making it impossible to exploit the near-field interference to achieve directional properties.” To me the structure shown in this manuscript is exactly “the near-field interference” of “chiral dipolar sources.”
4. Regarding the experimental part, this paper (DOI: <https://doi.org/10.1021/acsnano.9b04046>) experimentally studied a similar structure and claimed that the g -factor, representing the degree of asymmetry of chiral light scattering of single nanoparticles, can reach up to 0.8, which is higher than what was reported here. Could the authors explain what is unique about their findings in this manuscript compared to that paper other than asymmetric scattering?
5. In Figure 3g, could the authors clarify why the g -factor changes sign near 680 nm, given that the intensity of RCP is consistently larger than that of LCP? Additionally, why is the g -factor suspiciously similar to that in Figure 3f, even though the intensity of LCP and RCP is completely opposite?
6. The authors have demonstrated that the relative difference in the scattered intensities of LCP and RCP light in the zoomed-in area corresponds to 15% in Figure 4. It would be more persuasive if the authors could provide an estimate of the power of the scattered circularly polarized light and a numerical estimate of the scattering efficiency of these nanohelicoids. Furthermore, comparing this efficiency with that of other chiral nanoparticles would help to place this manuscript in the proper context.

Minor comments:

1. It would be beneficial if the authors could enhance the clarity of Figure 2d, 3b and 3c. There are too many curves so it's

not obvious which data is supportive of the main claim.

2. The coordinate system should also be indicated in Figure 2a and 4a.

3. Please add some descriptive text or symbols to Figure 4 b-i for readers to follow the meaning of the panels.

4. It would be helpful if the authors could explain how they isolate single L-nanohelicoids from the ensemble in the methods.

Reviewer #2

(Remarks to the Author)

Reviewer #3

(Remarks to the Author)

In this manuscript, the authors propose the concept of rotating chiral dipoles to engineer the direction of chiral scattering. Under the circularly polarized light illumination, rotating chiral dipoles are excited in the nanoparticle. The authors investigate the magnitudes and phases of chiral dipoles and far-field scattering intensities with both LCP and RCP illumination numerically. The authors experimentally implement the directional chiral scattering from plasmonic nanohelicoidal particles by imaging the Fourier plane. The intensity of scattering is enhanced when the handedness of illuminated light is the same as that of nanohelicoidal particles. The manuscript is clear and the supporting materials provide additional discussion for rotating chiral dipoles. The concept is interesting but there are several major concerns that should be addressed.

1. The importance and necessity of unidirectional scattering are not emphasized in this manuscript. Although the introduction of rotating chiral dipoles is to manipulate the direction of chiral scattering, how direction control can be useful in applications. Considering the readers of Nature Communications, the authors should add comprehensive explanations about the applications of chiral light-matter interaction. Sufficient discussion is needed about applications such as chiral sensing and quantum optical technology mentioned in the manuscript. Additional experiments such as biosensing using chirality should be included to prove the usefulness of directional chiral scattering.

2. In the manuscript, a detailed analysis and experiment on unidirectional chiral scattering are conducted. I recommend authors attach the advantages of this unidirectional chiral response compared to cases where there is no directionality, as shown in Fig. 1a and 1b. Quantitatively, how much does the chirality differ between these two scenarios? It is also recommended to demonstrate experimentally how the response differs in these cases. Furthermore, a more detailed explanation of the novelty when the chiral response is directional would be valuable. As referenced in the manuscript, many studies have investigated the chiral response using similar chiral particles. I suggest highlighting the novelty of this study in comparison to the previous works.

3. Generally, nanoparticle scattering is measured when the nanoparticle is placed on a substrate. The presence of the substrate can alter the optical response. Here, it seems that there is no need to consider the substrate due to the assumption that the particle is embedded in a SiO₂ matrix. However, how significantly does the presence of a substrate affect chiral scattering? I recommend including content regarding the scattering response with and without the presence of a substrate.

4. In the manuscript, the scattering of L-nanohelicoid is calculated and the corresponding experimental results are presented. Then, does D-nanohelicoid, which has opposite helicity, show the expected results? I recommend experimentally and theoretically demonstrating the scattering response of these two opposite types of nanoparticles.

Reviewer #4

(Remarks to the Author)

An interesting manuscript from excellent Zayats' research group. The manuscript provides theoretical justification and experimental confirmation of the polarization-dependent scattering of RCP or LCP light from chiral Au helicoids. The scattering was founded to be a function of NPs and light chirality. This is undoubtedly an interesting work that deserves publication in Nature Comm. However, I have a few points that the authors should consider.

Will light scattering change with a change in the position (orientation) of the nanoparticle relative to the incident light beam?

In the case of experimental results, can the authors provide SEM that will show the distribution of nanoparticles on the substrate? Whether scattering was measured from 'individual' nanoparticle(s) or from their self-assembled structure (which is less likely, since glutathione-prepared helicoids generally do not have a pronounced tendency to self-assemble).

It would also be good to describe in a little more detail the part related to the potential application of the results obtained in practical areas. For example, it is quite clear how the results obtained can be used in the field of sensors and optomechanics. However, their application in the quantum optical field remains unclear (chiral luminescence?). It would also be good if the authors confirmed the potential application with references from the literature (few papers describing the utilization of chiral plasmon-active nanostructures in sensing or CPL production have already been published).

Reviewer #5

(Remarks to the Author)

The paper by Yuanyang et al. describes a new nanophotonic concept, namely the concept of a spinning chiral dipole scatterer. This should be contrasted to earlier reports on spinning dipoles (spinning electric-only dipoles), and chiral dipoles

(aligned, but quarter-wave shifted electric and magnetic dipoles), and fall in the wider class of Mie scattering phenomena discussed in the metasurfaces literature. The claim is that these spinning dipoles can be employed to obtain directional scattering as well as selective radiation of circular polarization that matches the handedness of the chiral dipole. As such, this work will appeal both to the metasurfaces community and to the community interested in superchiral light, chiral scattering, and chiroptical effects. The authors supplement the predictions with experiments, and demonstrates this dominating forward scattering of circularly polarized light by single chiral helicoids using dark-field Fourier microscopy. These are hard experiments, show compelling evidence, and the work is in good agreement with the numerically calculated polarization-resolved radiation patterns. Overall I believe this work is very solid, and could appeal to the nanophotonics community as described above. It would certainly fit in a top tier journal such as Light Sci Appl, Nanophotonics, Nano letters or similar. Whether that is at the level of Nature Commun I am not sure.

Overall the work is very clear and there is little to improve. Some specific points:

1. The term "rotating chiral dipoles" that this paper introduced is a novel term to many readers, even for just rotating electric dipoles. An even more didactic introduction or graphical impression of electric dipoles, magnetic dipoles, and spinning versions thereof, is in my view useful and required even before coming to Figure 1. As is, Figure 1 is quite technical and maybe very hard for the broad audience of Nature Commun.
2. The authors stated that the concept "opens up numerous possibilities for engineering of scattering from chiral nanostructures and optical nano-antennas for the design and application of chiral light-matter interaction". Perhaps it is better to support this statement with some corresponding descriptions or examples. Without these, this is a rather empty statement.
3. I have a particular question about normalization in Fig. 4. The authors plot S_3/S_0 , which is problematic at points of zero intensity $S_0=0$. The S_0 maps do appear to show zero intensity, and it is unclear how the authors deal with normalization by 0 and the associated issues with noise, and diverging numbers.
4. In figure 4a, I assume there is air between the sample and the 0.95 NA objective? If that is the case, the forwardly scattered light should come out at angle greater than the Brewster angle, since the illumination angle is 65 degrees. This should affect the polarization of the radiation pattern, perhaps the authors should comment on whether this, or more generally the Fresnel coefficients of the substrate, affect their main conclusions.

Version 1:

Reviewer comments:

Reviewer #1

(Remarks to the Author)

The authors have addressed our comments, so we recommend it for publication.

Reviewer #2

(Remarks to the Author)

Reviewer #3

(Remarks to the Author)

In this work, the authors introducing directional chiral scattering using the rotating chiral dipoles. The authors extend the discussion of directional control in the manuscript to emphasize the importance of unidirectional chiral scattering from plasmonic nanoparticles. A detailed explanation of chirality in the nanoparticle's response is provided. Numerical simulations are performed to investigate the influence of the substrate on the chiral scattering. The authors include the discussion about how the permittivity of the substrate affects the chiral scattering response in the manuscript. According to the additional simulation for the scattering of D-nanohelicoids shown in Fig. R3, their scattering response is the reverse of L-nanohelicoids as expected. Overall, the manuscript includes a more comprehensive explanation of the concept and objectives of this work. The previous concerns have been well addressed, and the manuscript has been significantly improved. Therefore, I recommend it for the publication in Nature Communications.

Reviewer #4

(Remarks to the Author)

The authors performed a careful revision of the manuscript and have answered all questions and comments satisfactorily. The manuscript can be published in its current form.

Dear Editor,

Thank you very much for sending us the Reviewers' comments on our manuscript. We are very grateful to all the Reviewers for their careful reading, thoughtful recommendations and high opinion about our paper. We have taken all the comments into account in the revised version of the manuscript as follows. We hope very much that the revised version is suitable for publication your journal.

Best regards,
Yuanyang Xie, Anatoly Zayats
Corresponding authors

RESPONSE TO REVIEWERS' COMMENTS

Reviewer #1&2

Comment 1. *In terms of the "rotating chiral dipoles" concept, there is a statement "The basis of the four rotating chiral dipoles forms a complete set describing any dipolar source arbitrarily oriented in space." This is clearly inaccurate since SI showed the set can only represent p_x , p_y , m_x , m_y , not p_z and m_z . I believe the set can be completed by adding two more rotating chiral dipoles, but it will probably destroy the elegant symmetry presented in the current set.*

RESPONSE. We are very grateful to the Reviewers for the important comment and noticing the inconsistency. Indeed, four rotating chiral dipoles can represent the dipolar excitations in the xy -plane, while six rotating dipoles form a complete set in three dimensions. We have clarified this in the revised versions of the main text (page 6, lines 1–4) and Supplementary Information (page 2, Supplementary Section 1).

Comment 2. *Now if we ignore p_z and m_z , then there is no fundamental difference between the "rotating chiral dipole" set and the linear p and m set. This is not a problem since the latter is known to be complete, since the introduction of a new set would still be justified if the authors can offer a real physical object that is at least theoretically a pure base in the new set. Ideally, there should be some simple symmetry argument showing this object must be a pure rotating chiral dipole. Alternatively, there is some new property demonstrated by the pure rotating chiral dipole. The structure studied in this manuscript satisfied neither. As the manuscript introduced, directional circularly polarized scattering can be engineered using other frameworks too. It doesn't present any physical insight if one just defines an arbitrary orthogonal basis set, and for an arbitrary structure finds coincidentally a condition that makes the optical response similar to one of the bases.*

RESPONSE. We thank the Reviewers for the comment and very sorry that we were not clear in this respect. We agree with the Reviewers that some objects are commonly associated with given plasmonic resonances, but even a deeply-subwavelength spherical metallic nanoparticle, which is considered an archetypic object for a dipolar plasmonic resonance also support all other higher-order modes, which are weakly coupled to light, but still exist with their own resonant frequencies, so it cannot be a pure base in this case. Indeed, any plasmonic object possess a family of plasmonic

resonances within a given set, at certain resonant frequencies. In this respect, rotating chiral dipoles satisfy all the conditions which were required for the introduction of any other set for dipolar excitations: 1) pure chiral dipolar states can be excited, e.g., in the nanohelicoid at a wavelength of 620 nm (as in this work), and indeed in other plasmonic nanoparticles satisfying the resonant conditions, with other plasmonic modes existing at other wavelengths; 2) any dipolar plasmonic excitation can be decomposed into the set of rotating chiral dipoles, because they produce a complete set, as was rightly pointed out by the Reviewers. The Reviewers are absolutely right that there is a particular new property associated with them to justify their introduction, and as we have demonstrated in the manuscript this new property is unidirectional chiral scattering. The proper choice of an appropriate complete set based on rotating chiral dipoles allows to understand physics behind the problem and make physics-informed predictions. In order to clarify these issues, we added a short discussion on page 11, lines 19–23, of the revised manuscript.

Comment 3. *Another vague statement is “In contrast, for chiral dipolar sources, in the radiation patterns of the involved collinear magnetic and electric dipoles, the respective electric fields are orthogonal to each other and do not interfere, making it impossible to exploit the near-field interference to achieve directional properties.” To me the structure shown in this manuscript is exactly “the near-field interference” of “chiral dipolar sources.”*

RESPONSE. We are sorry for the confusion. At the point at which this sentence appears in the introduction, the concept of rotating chiral dipoles has not been introduced yet, so this sentence describes the situation of a **non-rotating** chiral dipolar source, where there is no near-field interference, as the fields are orthogonal to each other. To make absolutely clear which type of the dipolar sources is meant, in the revised manuscript we have added the reference on non-rotating chiral dipoles at the end of this sentence. Indeed, the premise of this manuscript is that when we made the chiral dipoles **rotating** in order to achieve the near-field interference leading to the unidirectional chiral scattering. We have checked that this notion is clear in the revised manuscript.

Comment 4. *Regarding the experimental part, this paper (DOI: <https://doi.org/10.1021/acsnano.9b04046>) experimentally studied a similar structure and claimed that the g-factor, representing the degree of asymmetry of chiral light scattering of single nanoparticles, can reach up to 0.8, which is higher than what was reported here. Could the authors explain what is unique about their findings in this manuscript compared to that paper other than asymmetric scattering?*

RESPONSE. We thank the Reviewers for the valuable comment. The difference comes from the following factors. Firstly, the wet-chemical fabrication brings a slight difference between different nanostructures fabricated even in the same batch, notwithstanding different batches. Secondly, one cannot compare g-factors of the nanoparticles in different environments, as the refractive index of surroundings has strong influence on the modes of the nanoparticle (as described on page 13, lines 9–14, of the revised manuscript). Therefore, one cannot compare g-factors measured in dissimilar conditions. Indeed, our manuscript is not about fabrication of the specific chiral nanoparticles but demonstration of a new general concept of the rotating chiral dipoles, which can be applied to various

nanoparticles; we experimentally demonstrate it on the example of nanohelicoids exactly for the reason that their other properties are well known. Indeed, the demonstration of unidirectional scattering is a point of this paper.

Comment 5. *In Figure 3g, could the authors clarify why the g-factor changes sign near 680 nm, given that the intensity of RCP is consistently larger than that of LCP? Additionally, why is the g-factor suspiciously similar to that in Figure 3f, even though the intensity of LCP and RCP is completely opposite?*

RESPONSE. We thank the Reviewers for noticing this and sorry for misunderstanding. The g-factors in Figure 3f and 3g were **the same** (see the red lines in the plots in the original figures). The g-factor was calculated using the scattering **cross-sections for LCP and RCP light illuminations**, which enter the same formula (Eq. 2 in the manuscript). Orange and green lines in those figures correspond the scattered light intensity (**NOT** illumination light polarisation). In order to avoid the confusion, in the revised manuscript we have separated plots of the g-factor and the scattered intensity in the revised Fig. 3.

Comment 6. *The authors have demonstrated that the relative difference in the scattered intensities of LCP and RCP light in the zoomed-in area corresponds to 15% in Figure 4. It would be more persuasive if the authors could provide an estimate of the power of the scattered circularly polarized light and a numerical estimate of the scattering efficiency of these nanohelicoids. Furthermore, comparing this efficiency with that of other chiral nanoparticles would help to place this manuscript in the proper context.*

RESPONSE. We thank the Reviewers for the valuable suggestions. The scattering efficiencies of the nanohelicoids for LCP and RCP illuminations have been added as a new Supplementary Fig. 9 in the revised Supplementary Information. They were further compared with a helix nanoparticle, which is an archetypical chiral object.

Fig. R1. (a) The simulated scattering efficiency for left-handed gold helix and helicoid nanoparticles of comparable sizes in SiO₂. (b, c) Far-field scattering intensity and polarisation diagrams of the right-handed helix under 620 nm (b) LCP and (c) RCP illumination.

Comment 7. *It would be beneficial if the authors could enhance the clarity of Figure 2d, 3b and 3c. There are too many curves so it's not obvious which data is supportive of the main claim.*

RESPONSE. We thank the Reviewers very much for the suggestions. We have replotted Figs. 2 and 3 in the revised manuscript in order to enhance the clarity. More information has also been added in the figure caption. We understand that Figs. 3b and 3c present several curves but it is very important to be able to compare these curves, particularly the amplitudes and phases of the excited dipoles, in order to understand the nanohelicoid dipolar response.

Comment 8. *The coordinate system should also be indicated in Figure 2a and 4a.*

RESPONSE. We thank the Reviewers for the suggestion. A coordinate system has been added to Fig. 4a of the revised manuscript. Also, in the caption of Fig. 2a it has been added that the SEMs show the top view of the nanohelicoids on a substrate, which explicitly defines the spatial position of the structures.

Comment 9. *Please add some descriptive text or symbols to Figure 4 b-i for readers to follow the meaning of the panels.*

RESPONSE. We thank the Reviewers for the valuable comment. To increase clarity of Fig. 4b-i labels “LCP” and “RCP” with the subscripts “exp” or “sim” have been added to clearly identify the Fourier-space maps.

Comment 10. *It would be helpful if the authors could explain how they isolate single L-nanohelicoids from the ensemble in the methods.*

RESPONSE. We thank the Reviewers for the valuable comment. The particles were separated on a substrate by spin coating. This has been clarified in the revised manuscript (page 11, lines 2–5 from the bottom) and a SEM image (Supplementary Fig. 13) showing that after the spin coating the nanohelicoids are well-separated has been added to the revised Supplementary Information.

Reviewer #3

Comment 1. *The importance and necessity of unidirectional scattering are not emphasized in this manuscript. Although the introduction of rotating chiral dipoles is to manipulate the direction of chiral scattering, how direction control can be useful in applications. Considering the readers of Nature Communications, the authors should add comprehensive explanations about the applications of chiral light-matter interaction. Sufficient discussion is needed about applications such as chiral sensing and quantum optical technology mentioned in the manuscript. Additional experiments such as biosensing using chirality should be included to prove the usefulness of directional chiral scattering.*

RESPONSE. We thank the Reviewer for the valuable comment. Extended discussion related the directional control and chiral light-matter interaction have been added on pages 2–3 and page 14 of the revised manuscript. As far as additional experiments are concerned, this paper is about proposing a concept of the rotating chiral dipoles and demonstrating unidirectional chiral scattering on its basis rather than developing particular applications mentioned above, each of which would constitute a separate project.

Comment 2. *In the manuscript, a detailed analysis and experiment on unidirectional chiral scattering are conducted. I recommend authors attach the advantages of this unidirectional chiral response compared to cases where there is no directionality, as shown in Fig. 1a and 1b. Quantitatively, how much does the chirality differ between these two scenarios? It is also recommended to demonstrate experimentally how the response differs in these cases. Furthermore, a more detailed explanation of the novelty when the chiral response is directional would be valuable. As referenced in the manuscript, many studies have investigated the chiral response using similar chiral particles. I suggest highlighting the novelty of this study in comparison to the previous works.*

RESPONSE. We thank the Reviewer for the valuable comment. The advantages of unidirectional chiral response have been added on pages 2–3 of the revised manuscript. Chirality of the nanoparticle response is determined by the presence and strength of the aligned and spectrally overlapped electric and magnetic dipoles (i.e., the strength of a resulting chiral dipole). As for the chirality of light, in the case of the excitation of pure chiral dipoles shown in Fig. 1, it does not depend on whether the dipole is a conventional one or rotating, it is either pure RCP or pure LCP, but the directionality is different in these two cases. We have added this important point of explanation in the revised manuscript on page 11, lines 14–19. The excitation of conventional (non-rotating) chiral dipoles for the comparison requires the use of nanoparticles different from nanohelicoids and was extensively studied (see e.g. Ref. 6 cited in the manuscript), and thus is beyond the scope of this manuscript. As the Reviewer mentioned that many properties of the nanohelicoids were studied in the literature. This manuscript uses helicoids only as an example of a nanoparticle where rotating chiral dipoles can be excited for exactly this reason. The demonstration of a new concept of the rotating chiral dipoles is the objective of the manuscript, which was demonstrated on the example of a nanohelicoid. We have checked that the objective of the study is clear throughout the revised manuscript.

Comment 3. *Generally, nanoparticle scattering is measured when the nanoparticle is placed on a substrate. The presence of the substrate can alter the optical response. Here, it seems that there is no need to consider the substrate due to the assumption that the particle is embedded in a SiO₂ matrix. However, how significantly does the presence of a substrate affect chiral scattering? I recommend including content regarding the scattering response with and without the presence of a substrate.*

RESPONSE. We thank the Reviewer for the insightful comment. We have performed additional numerical simulations to check the influence of a substrate on the scattering from the nanohelicoid (Fig. R2). Despite a helicoid in air has a much weaker chiral response than in water or SiO₂, as expected due the change in the refractive index, the key effect of the forward chiral enantio-sensitive scattering can be still clearly observed in the presence of a substrate; although the influence from the substrate does exist, as seen in the broadening of the scattering diagrammed features.

Fig. R2. Scattering of (a) LCP and (b) RCP light incident at 65 degrees on a helicoid, placed in a uniform matrix or on a substrate made from various media. In the substrate case the fields were analysed at a distance far enough from the nanohelicoid, so that they are converged to the far-field scattering maps.

A discussion on the influence of the variation of the permittivity of surroundings and/or the presence of a substrate with dissimilar optical properties has been added to the revised manuscript (page 13, lines 9–14), while Fig. R2 has been added as Supplementary Fig. 15 in the revised Supplementary information.

Comment 4. *In the manuscript, the scattering of L-nanohelicoid is calculated and the corresponding experimental results are presented. Then, does D-nanohelicoid, which has opposite helicity, show the expected results? I recommend experimentally and theoretically demonstrating the scattering response of these two opposite types of nanoparticles.*

RESPONSE. We thank the Reviewer for the important comment. We indeed checked the optical response of D-helicoids (Fig. R3), they do behave as should be expected, i.e. reversely to L-helicoids. We have included this observation in the revised version of the manuscript (page 11, lines 11–14) and Fig. R3 has been added to the revised Supplementary information as Supplementary Fig. 12. Experimentally, for the fabrication of D-nanohelicoids, D-glutathione is required. As it is a much rarely used chemical, the available purity of it is much worse compared to L- glutathione. Therefore, it is impossible to fabricate D-nanohelicoids with a comparable quality to that of L-nanohelicoids, which hinders their objective comparison. Since the focus of this manuscript is on the properties of rotating

chiral dipoles, additional, but not completely fair comparison of the nanohelicoid handednesses in the experimental observation would overcrowd the presentation.

Fig. R3. Far-field scattering intensity and polarisation diagrams of a D-helicoid calculated under (a) LCP and (b) RCP 620-nm plane wave illumination. The colour indicates the polarisation state of the scattered light calculated from Stokes parameters S_3/S_0 (1: RCP; -1: LCP).

Reviewer #4

Comment 1. *Will light scattering change with a change in the position (orientation) of the nanoparticle relative to the incident light beam?*

RESPONSE. We thank the Reviewer for the important comment. Yes, based on the simulations presented in Supplementary Fig. 4, the direction of incident light has an impact on the scattering from a chiral nanoparticle as the magnitudes of magnetic and electric dipoles as well as the phase between them will slightly change. Nevertheless, the unidirectional character of scattering and the pure handedness of the scattered light do not change significantly. We have clarified the related discussion in the revised main text (page 13, lines 2–9).

Comment 2. *In the case of experimental results, can the authors provide SEM that will show the distribution of nanoparticles on the substrate? Whether scattering was measured from ‘individual’ nanoparticle(s) or from their self-assembled structure (which is less likely, since glutathione-prepared helicoids generally do not have a pronounced tendency to self-assemble).*

RESPONSE. We thank the Reviewer for the valuable comment. In the revised Supplementary Information we have added a SEM image showing the distribution of nanohelicoids on the substrate after the spin coating (Supplementary Fig. 13) and made a reference to it in the revised main text (page 11, lines 2–5 from the bottom).

Comment 3. *It would also be good to describe in a little more detail the part related to the potential application of the results obtained in practical areas. For example, it is quite clear how the results obtained can be used in the field of sensors and opto-mechanics. However, their application in the quantum optical field remains unclear (chiral luminescence?). It would also be good if the authors confirmed the potential application with references from the literature (few papers describing the utilization of chiral plasmon-active nanostructures in sensing or CPL production have already been published).*

RESPONSE. We thank the Reviewers for the valuable suggestion. An extended discussion related to the potential application of the directional chiral scattering together with relevant references has been added on of the revised manuscript in the introduction. More referenced examples, particularly about the applications in opto-mechanics, sensing and chiral luminescence have been given in the Discussion section.

Reviewer #5

Comment 1. *The term “rotating chiral dipoles” that this paper introduced is a novel term to many readers, even for just rotating electric dipoles. An even more didactic introduction or graphical impression of electric dipoles, magnetic dipoles, and spinning versions thereof, is in my view useful and required even before coming to Figure 1. As is, Figure 1 is quite technical and maybe very hard for the broad audience of Nature Commun.*

RESPONSE. We thank the Reviewer for the valuable comment. In the revised Supplementary Information, we have added a new Supplementary Fig. 6 graphically showing schematics for electric, magnetic and chiral dipoles, along with their rotating counterparts, accompanied with the corresponding far-field intensity/polarisation diagrams and near-field radiation patterns. We have also added references to this figure in the relevant places of the revised manuscript (page 4).

Comment 2. *The authors stated that the concept “ opens up numerous possibilities for engineering of scattering from chiral nanostructures and optical nano-antennas for the design and application of chiral light-matter interaction”. Perhaps it is better to support this statement with some corresponding descriptions or examples. Without these, this is a rather empty statement.*

RESPONSE. We thank the Reviewer for the valuable comment and sorry we have not done it in the original version. Extended discussion related to the potential applications has been added on pages 2 and 3 of the revised manuscript together with relevant references. More referenced examples, particularly about the applications in opto-mechanics, sensing and chiral luminescence have been given in the Discussion section.

Comment 3. *I have a particular question about normalization in Fig. 4. The authors plot S_3/S_0 , which is problematic at points of zero intensity $S_0=0$. The S_0 maps do appear to show zero intensity, and it is*

unclear how the authors deal with normalization by 0 and the associated issues with noise, and diverging numbers.

RESPONSE. We are grateful for this important comment requiring clarifications. While the normalisation of Stokes parameters by S_0 is usual procedure¹, it would indeed be problematic in the points where the intensity is close to zero. As it follows from the obtained results, in both experiments and numerical simulations the scattering intensity in any direction was much larger than the signal noise (numerical noise in the case of simulations), so the above problem was not met. We emphasised this in the caption of Fig. 3 in the revised manuscript.

1. Bohren, C. F., & Huffman, D. R. (2008). Absorption and scattering of light by small particles. John Wiley & Sons.

Comment 4. *In figure 4a, I assume there is air between the sample and the 0.95 NA objective? If that is the case, the forwardly scattered light should come out at angle greater than the Brewster angle, since the illumination angle is 65 degrees. This should affect the polarization of the radiation pattern, perhaps the authors should comment on whether this, or more generally the Fresnel coefficients of the substrate, affect their main conclusions.*

RESPONSE. We thank the Reviewer for the insightful comment. Considering the change of polarisation in the collection path, for the high NA used in the experiments the maximum collection angle is 71.8 degrees. For this collection angle, marking the largest degree of the polarisation change, the difference in the amplitude transmission of s- and p-polarised components of forward-scattered circularly polarised light on the glass/air interface is 14.5%. This, seemingly large difference leads to only 0.012 change of the polarisation $S_3/S_0 = -2 \cdot \text{Im}(E_s E_p^*) / (|E_s|^2 + |E_p|^2) = -2 \cdot \text{Im}(1 \cdot (+/- 0.855i)^*) / (|1|^2 + |0.855|^2) = 0.988$, compared to the incident light $S_3/S_0 = 1$. Thus, the changes of the polarisation at the glass/air interface are negligible. We have included this discussion in the revised manuscript (page 15, lines 16–19).